# Attachments for the Orthodontic Aligner Treatment—State of the Art—A Comprehensive Systematic Review

**DOI:** 10.3390/ijerph20054481

**Published:** 2023-03-02

**Authors:** Maciej Jedliński, Marta Mazur, Mario Greco, Joyce Belfus, Katarzyna Grocholewicz, Joanna Janiszewska-Olszowska

**Affiliations:** 1Department of Interdisciplinary Dentistry, Pomeranian Medical University in Szczecin, 70-111 Szczecin, Poland; 2Department of Dental and Maxillofacial Sciences, Sapienza University of Rome, 00161 Rome, Italy; 3Department of Paediatric Dentistry, University of L’Aquila, 67100 L’Aquila, Italy; 4Faculty of Dentistry, Universidad de los Andes, Santiago 7620001, Chile

**Keywords:** aligners, attachments, positioning, clinical effectiveness, position, tooth movement, orthodontics, CAD

## Abstract

Background: In recent years the burden of aligner treatment has been growing. However, the sole use of aligners is characterized by limitations; thus attachments are bonded to the teeth to improve aligner retention and tooth movement. Nevertheless, it is often still a challenge to clinically achieve the planned movement. Thus, the aim of this study is to discuss the evidence of the shape, placement and bonding of composite attachments. Methods: A query was carried out in six databases on 10 December 2022 using the search string (“orthodontics” OR “malocclusion” OR “Tooth movement techniques AND (“aligner*” OR “thermoformed splints” OR “invisible splint*” AND (“attachment*” OR “accessor*” OR “auxill*” AND “position*”). Results: There were 209 potential articles identified. Finally, twenty-six articles were included. Four referred to attachment bonding, and twenty-two comprised the influence of composite attachment on movement efficacy. Quality assessment tools were used according to the study type. Conclusions: The use of attachments significantly improves the expression of orthodontic movement and aligner retention. It is possible to indicate sites on the teeth where attachments have a better effect on tooth movement and to assess which attachments facilitate movement. The research received no external funding. The PROSPERO database number is CRD42022383276.

## 1. Introduction

In recent years, the burden of aligner treatment has been growing, as thermoformed splints are highly appreciated by patients due to the improved aesthetics and greater comfort with respect to traditional fixed orthodontic appliances [1,2]. However, the sole use of aligners without auxiliaries is characterized by a series of limitations as they can only push teeth into preformed spaces away from the center of resistance [3]. Interest from patients has encouraged manufacturers to look for new solutions to improve the therapeutic characteristics of their products by introducing auxiliaries such as composite attachments, bite ramps, precise cuts, and power ridges, thus allowing the treatment of more types of malocclusions with aligners [4]. Thanks to the use of composite attachments, tooth movement can be better controlled and actively guided, increasing the contact area and locating the point of force application closer to the center of resistance, thus allowing a more bodily tooth movement [5]. Thus, composite attachments began to be an integral part of the aligner treatment. Nevertheless, it is often still a challenge to clinically achieve the planned teeth movement [6]. Thus, the authors of the present study aimed to find scientific evidence referring to the use of attachments, especially their optimal placement for efficient tooth movement, as well as their number and shape. The aim of this systematic review is to comprehensively discuss the scientific evidence on how composite attachments should be bonded to achieve the best possible treatment results in aligner therapy.

## 2. Materials and Methods

This systematic review was undertaken according to the PRISMA statement [7] and PRISMA reporting guidelines [8,9], together with the indications from the Cochrane Handbook for Systematic Reviews of Interventions [10]. The framework of this systematic review, according to PICOS [11], was as follows. Population: clear orthodontic aligners; Intervention: attachment bonding; Comparison: different placement of the composite attachments; OR different composites used to make the composite attachments; Outcomes: efficiency of movement and success rate; Studies: in vitro studies, retrospective clinical studies and prospective clinical studies. The PICO questions were as follows: How should an orthodontist place the attachment to induce the orthodontic movement efficiently? What shape of the attachment should an orthodontist choose to induce orthodontic movement efficiently? What materials should be chosen to induce orthodontic movement efficiently?

### 2.1. Search Strategy

Literature searches of free text and MeSH terms were performed by using MedLine (PubMed), Scopus, Web of Science and Embase (from 1950 to 10 December 2022). All of the searches were performed using a combination of subject headings and free-text terms. The final search strategy was determined through several pre-searches. The keywords used in the search strategy were as follows: (“intraoral scanners AND efficiency AND accuracy AND orthodontics”). The search strategies used for the different search engines were as follows:-For MedLine (PubMed) and PubMedCentral: (“orthodontics”[MeSH terms] OR “malocclusion”[MeSH Terms] OR “Tooth movement techniques”[Mesh Major Topic]) AND (“aligner*”[All fields] OR “thermoformed splints”[All fields] OR “invisible splint*”[All fields]) AND (“attachment*”[All fields] OR “accessor*”[All fields] OR “auxill*”[All fields] AND “position*”[All fields]);-For Scopus: TITLE-ABS-KEY (((orthodontic*) OR (malocclusion)) AND (aligner*) AND ((attachment*) OR (accessor*) OR (auxill*) AND position*);-For Web of Science (“orthodontics” OR “malocclusion”) AND (“aligner*”OR “thermoformed splints” OR “invisible splint*”) AND (“attachment*” OR “accessor*” OR “auxill*” AND “position*”) (All Fields) and Article (Document Types);-For Embase: (“orthodontics” OR “malocclusion”) AND (“aligner*”OR “thermoformed splints” OR “invisible splint*”) AND (“attachment*” OR “accessor*” OR “auxill*” AND “position*”);-For EBSCO Dental and Oral Sciences: (“orthodontics” OR “malocclusion”) AND (“aligner*”OR “thermoformed splints” OR “invisible splint*”) AND (“attachment*” OR “accessor*” OR “auxill*” AND “position*”).

The reference lists of the primary research reports were cross-checked in an attempt to identify additional studies. However, no additional studies were found or added. The study protocol was registered in the PROSPERO database with the number CRD42022383276.

### 2.2. Eligibility Criteria

The following inclusion criteria were employed for this systematic review: (1) prospective clinical trial; (2) finite element analysis (FEM); (3) retrospective clinical trials; (4) and in vitro studies (5) studies published in English. All of the potentially evaluated articles were supposed to explore the subject of bonding and the placement of composite attachments for clear aligner treatments to achieve superior treatment results.

The following exclusion criteria were applied: (1) case reports; (2) reviews; (3) abstract and author debates or editorials; (4) a lack of effective statistical analysis; (5) papers related to different aspects of the biomechanics of aligner treatment; (6) and grey literature (unpublished/unreviewed results).

### 2.3. Data Extraction

The titles and abstracts were studied separately by two researchers (MJ and MM), searching for studies compliant with the inclusion criteria using Zotero reference manager. Suitability for inclusion was assessed through full-text reading. Disagreements were discussed with the third author (JJO). However, the agreement between the authors was high and yielded a Cohen kappa coefficient of 0.99. Authorship, year of publication, type of each eligible study and its relevance regarding the placement and preparation of composite attachments were extracted by the first author (MJ) and examined by the second and third authors (JJO and MM). The characteristics of the studies included are presented in Table 1.

### 2.4. Quality Assessment

According to the PRISMA statements, the evaluation of methodological quality provides an indication of the strength of the evidence provided by the study because methodological flaws can result in biases [6] (Supplementary File S1 and S2). As there were 4 types of studies included, each type was evaluated with a type-specific scale (randomized clinical trials, case–control studies, FEM studies and in vitro studies).

The revised tool for assessing the risk of bias in randomized trials (RoB 2) was used to perform the quality assessment of the randomized clinical trials [12]. In order to find out the level of the risk of bias, it was considered in the assessment whether the study was randomized, whether the subject was correctly allocated and blinded with appropriately described methods and how the outcome was presented. There are three possible grades for each characteristic: low RoB, meaning no bias, or if present, rather unlikely to alter the results significantly; some concerns, meaning a risk of bias that raises some doubt about the results; and high RoB, a bias that may alter the results significantly. Based on each of the five characteristics, the final score is given. Moreover, for Cross-sectional Studies, the Newcastle–Ottawa Quality Assessment Form [13] was used. The quality assessment of all the included case–control studies was based on object selection, comparability and exposure. The possible quality assessment score ranged from zero to nine points, with a high score indicating a good quality study. There was at least one point (star) awarded for each characteristic evaluated. For comparability, outcome criteria and ascertainment of exposure, the study could have received two stars. Finite element analyses were assessed using Methodological Quality Assessment of Single-Subject Finite Element Analysis Used in Computational Orthopedics (MQSSFE)—a specialized tool used for studies concerning biomechanics. Two independent researchers used the 37 questions: the answers were YES/NO in case of absence/presence of risk of bias. In case of disagreement, the question scored 0.5 points [14]. In order to perform a quality assessment of dental in vitro studies, the QUIN assessment tool was used. In this case, two authors used the subsequent scoring system: (i) adequately specified (2-1 points); (ii) not specified (0 points); and not applicable (exclusion from the calculation). The overall score for the given research was calculated to classify the risk of bias (>70% = low, 50–70% = medium, and <50% = high) [15]. All of the specific criteria assessed using a given tool are provided in the respective tables (Table 2, Table 3, Table 4, Table 5 and Table 6).
ijerph-20-04481-t001_Table 1Table 1Included studies regarding bonding attachments.Authors and YearType of StudyStudied FeaturesSubjects within the StudyResults and Clinically Relevant ConclusionsAlsaud et al. 2022 [16]In vitro experimental studyThe bonding strength of attachments to ceramic restorationsIn total, 180 IPS e.max CAD specimens were divided into 12 different groups (*n* = 15 each). For surface roughness preparation, either 9.6% HF acid or 37% H_3_PO_4_ acid or air abrasion was used. For bonding, either Assure Bond or Single Bond Universal was used. The attachments were made either of condensable Filtek Z350 or of Z350 XT flowable composite.The highest shear bond strength of composite attachments to ceramic restorations was achieved by using HF acid or air abrasion with Assure Bond and Filtek Z350 condensable composite.Bruno et al. 2021 [17]Randomized control clinical trialThe influence of template type on bonding efficiency and attachment survival rateForty patients bonded with the Spark^®^ templateForty patients bonded with the Invisalign^®^ templateBoth groups had attachments made of Tetric EvoFlow^®^ Bulk Fill after a standard bonding procedureThe Spark group showed, in general, a lower frequency of debonding in comparison with the Invisalign group (87,5% vs. 73,5% of success rate after first bonding). Multiple failures occurred more often in the Invisalign group. However, no template characteristics proved to be crucial to this phenomenon.Chen et al. 2021 [18]In vitro experimental studyThe influence of composite material type on bonding efficiencyThe attachments were bonded with 3M Filtek Z350 condensable compositeThe attachments were bonded with 3M Filtek Z350 XT flowable compositeThe attachments were bonded with Sonicfill flowable compositeThe operation time of Z350XT Flowable and SonicFill was shorter than Z350XT.The shear bond strength was the highest for SonicFill.SEM showed that the bonding interface of Z350XT and SonicFill was compact.The 3D deviation and volumetric change in the 3D designed attachments and the attachments after actual bonding of Z350XT Flowable were greater than that of Z350XT and SonicFill.The wear volume loss of SonicFill and Z350XT was less than that of Z350XT Flowable.D’Antò et al. 2019 [19]In vitro experimental studyThe influence of composite material type on bonding efficiencyFlowable ENAMEL plus HRi^®^ Flow HF composite—70% fillerBracepaste^®^ orthodontic composite—72% fillerENAMEL plus HRi^®^ Enamel homogenous hybrid composite—80% fillerStudied composites of different viscosities and did not present any difference in the shape and volume of attachments reproduced with a template on extracted teeth. The orthodontic composite showed more overflow when compared to the flowable one.Valeri et al. 2022 [20]In vitro experimental studyThe influence of transfer tray design and composite viscosity on the accuracy of attachment bondingTransbond™ XT Light Cure Past attachments made with a rigid thicker transfer trayTransbond™ XT Light Cure Past made with a soft, thinner transfer trayTetricEvoflow made with a rigid thicker transfer trayTetricEvoflow made with a soft, thinner transfer trayTransbond™ XT Light Cure Past attachments made with a rigid thicker transfer tray are associated with higher accuracy and minor dispersion.

## 3. Results

### 3.1. Study Selection

There were 209 potential articles identified, from which 15 were found in PubMed, 53 in PubMed Central (PMC), 23 in Scopus, 45 in Web of Science, four in Embase and 69 in the EBSCO Dental & Oral Sciences database. After the removal of 83 duplicates, 126 titles and abstracts were assessed. Then, 85 papers were excluded, as they did not meet the inclusion criteria, being completely non-related to the topic of systematic reviews. The excluded studies mainly focused on other factors affecting the biomechanical performance of aligners, such as the accuracy of planned movement, splint thickness, wear length or material type. All of the remaining papers were retrieved. Of the remaining 41 papers, 15 were excluded because they were not relevant to the subject of the study. The resulting 26 papers were included in the qualitative synthesis. Four of them referred to attachment bonding procedures, and the other twenty-two comprised the influence of composite attachment on movement efficiency. A Prisma 2020 Flow Diagram representing the study selection is presented in Figure 1. (Figure 1 Flow diagram).

### 3.2. Study Characteristics

The included studies (Table 1) [16,17,18,19,20,21,22,23,24,25,26,27,28,29,30,31,32,33,34,35,36,37,38,39,40,41] were published between 2008 and 2022. However, the vast majority of them (twenty-one) are less than five years old. Out of the twenty-six studies, four groups of study types can be distinguished:In vitro studies(a)Finite element analysis;(b)In vitro experimental studies.In vivo studies(a)Cross-sectional studies;(b)Randomized clinical trials.

Considering the in vitro studies, there were eleven FEMs and eight experimental studies.

For the in vivo studies, the total sample size was 343 participants (mean: 49 per study). The mean age of the enrolled participants was impossible to define as half of the studies defined their participants as “non-growing patients or adult patients” without specifying the age range. However, in all studies, the patients were reported to be adult patients with full permanent dentition. A similar model (adult patient with full permanent dentition) was introduced into the simulation of all finite element analysis studies and in vitro experimental studies.

The study setting was as follows: (i) private clinic (4); (ii) hospital setting (1); (iii) and university setting (2).

None of the studies received any external funding. Three of the studies included detailed information related to the statal grant number.

The mean time of the declared follow-up in the clinical trials was 14,8 months.

The included studies were grouped into two main thematic areas:

(i) orthodontic bonding (total in vitro studies = four; total in vivo studies = one; total subjects, 80)

In four studies, the variable factor in the comparison was the physical characteristics of the composite. The main distinguishing feature was the amount of filler. In two studies, the variable factor was the different hardness and flexibility of the transfer tray. In the case of one study, the method used for tooth surface preparation was also a variable.

(ii) tooth movement (total in vitro studies = 15; total in vivo studies = six; total subjects = 243).

Nineteen studies included research related to the effect of the presence and positioning of the attachment on the teeth on movement efficiency. The comparators were different scenarios, such as the lack of an attachment and an attachment placed on a different side. Twelve of the included studies analyzed the shape of the attachment as a factor influencing the expression of tooth movement. Two of the studies included research on the shape and positioning of the attachment as a factor influencing aligner retention.

### 3.3. Risk of Bias Assessment

The risk of bias was assessed using proper quality assessment scales. Therefore, eleven studies [21,22,23,27,30,32,34,35,36,37,41] were evaluated with MSQFE. The study design and subject recruitment were in all FEMs, with most aspects covered in a manner consistent with the standards required. Only three studies [32,35,37] managed to perfectly configure and reconstruct a model according to the standards. In two studies [21,22], there were minor shortcomings in this regard. Eight studies [21,22,23,27,30,32,37,41] were characterized by proper boundary loading simulation. Five studies [21,22,30,35,37] perfectly validated the model, and two [23,36] were only characterized by minor shortcomings. Three studies perfectly described model assumption and validity according to the standards [22,30,35], and another three [21,32,37] were characterized by minor shortcomings. Eight studies [16,18,19,20,24,26,28,40] were subjected to analysis with the QUIN assessment tool. All of the in vitro studies clearly described the aim. Only three [16,18,19] performed a proper sample size calculation. In five studies, the sampling technique was clearly described, while in three, minor flow clouds were observed. Seven out of the eight studies [16,18,19,20,26,28,40] described the methodology in detail. Only four [19,20,28,40] of them properly described the operator. Five of included studies [16,18,19,24,40] described the data assessor details in accordance with the QUIN standards. None of the in vitro studies performed proper randomization and blinding. All of the in vitro studies included performed proper statistical analysis, and seven presented the results in detail [16,18,19,20,24,28,40]. From the RCTs included in the review, three performed [21,29,33] proper random sequence generation, and two [21,29] performed proper allocation concealment. In none of the included RCTs was it possible to maintain proper blinding at any stage of the trial. The methodology used in two of three studies included cross-sectional studies [31,39] that coped perfectly with the requirements listed in the Newcastle–Ottawa assessment form. In one of the cross-sectional studies, the proper sample size was not justified [38]. The risk of bias assessment is presented in the following tables; each study in the table is assigned to its study type (Table 2, Table 3, Table 4 and Table 5).

## 4. Discussion

This systematic review endeavored to comprehensively display the available evidence on the clinical application of composite attachment in aligner treatment. A total of 26 studies were included in this review, from which eleven were finite element analyses, eight were in vitro experimental studies, four were randomized clinical trials and three were cross-sectional studies. The included studies were critically revised, and they focused mainly on (i) the determination of the efficiency of the execution of digitally planned tooth displacement; (ii) attachment bonding procedures.

From the studies included, six FEMs [21,22,30,32,35,37], two in vitro studies [18,19], and two cross-sectional studies [31,39] were of low risk of bias in the light of quality assessment tools adapted to the types of research. Two RCTs were characterized as medium-quality evidence. The latter should be considered rather low-quality evidence. In high-risk-of-bias FEM studies, the most common shortcomings were a lack of model validation, a lack of boundary loading description or a lack of a description of the origin of the model used for the simulation. Many clinical and in vitro studies did not provide sample size adjustment and randomization. Due to the characteristics of the evaluated phenomena, full blinding was not possible in RCTs and in vitro studies because the aligners and composite attachments differed from one another, and an experienced researcher could easily recognize them with a high degree of probability. For the same reason, randomization would not bring much to the quality of the evidence coming from in vitro studies. In most cases, a common inspiration can be seen because later studies try not to repeat earlier methodological errors, even though the time distance is not significant.

It should be noted that attachment placement has two aspects: the first refers to the materials and procedures used for bonding, and the other to the shape and positioning of the attachments.

### 4.1. Attachment Bonding

First, it can be deduced from the included studies that harder, thicker transfer trays provide higher accuracy in transferring attachments to the teeth [17]. However, according to Bruno et al. [17], a less rigid Spark transfer tray was associated with easier tray removal and, thus, fewer attachment detachments during first-time bonding. However, this conclusion came from the tactile examination of the trays by the clinicians and objective measurements allowing a comparison of the physical characteristics between Spark and Invisalign trays were not made. For this reason, a clinician should choose more rigid trays, keeping in mind the fact that it may be more accurate but more difficult to perform bonding. From the studies dealing with the impact of composite viscosity on possible attachment clinical performance, there comes an easy dependency—the more filler, the better shear bond strength and better transfer accuracy. However, viscosity does not have an impact on the shape and volume of attachments. Condensable orthodontic composites are associated with longer chair time than flowable ones. Moreover, a condensable composite together with a precise, rigid transfer tray may be a challenge for the clinician to bond the attachments. Notwithstanding, in the D’Anto study, after exceeding 72% filler [19], accuracy and bonding strength did not increase significantly. A very promising material seems to be Kerr Sonicfill, which, thanks to its consistency, was easy to apply, but also retained bond strength and accuracy better than solid composites [18]. Therefore, based on the results of the included studies, it seems reasonable to recommend flowable composites or orthodontic bonding composites with as much filler as possible. Bonding attachments to ceramic restorations do not deviate from this principle, where prior chemical treatment is crucial—with HF acid or by air abrasion followed by Assure^®^ bond—avoiding standard bonds for composites [16]. It should be underlined that a better force transfer from the aligner to the tooth is associated with higher stress between the aligner and composite attachment with a tooth [20]. For this reason, it is imperative that properly designed attachments are as resistant to detachment as possible.

### 4.2. Tooth Movement

All of the studies included in the review indicate better control and a higher range of tooth movement in aligner treatment with attachments than without attachments, even if a given tooth is not to be moved but provides anchorage for movement [35]. This indicates that the use of attachments influences the success of the treatment. On the other side, aligner treatment with labial attachments is not fully invisible. However, as indicated by the studies included, in most cases, the attachments bonded to the palatal surfaces of the anterior teeth, providing a better expression of orthodontic movement than those bonded labially. Thus, the use of palatal attachments (if possible, in occlusion) may reduce the visibility of the treatment and, at the same time, improve its effectiveness. It should be kept in mind that palatal attachments have different biomechanical effects as they are positioned closer to the center of resistance. For many movements, it is the position of the attachment that is critical to avoid undesired movements to a similar or greater extent than the shape of the attachment. The clinically useful knowledge coming from the studies included is summarized in Table 7 in order to provide a clinical guideline. The attachments presented are those of the highest effectiveness according to the literature. It is noteworthy that in more difficult tooth movements (incisor extrusion, canine rotation and lateral segment distalization), it is advised to place attachments on the whole lateral segment—from the canine to the second molar (but especially important in adjacent teeth)— in order to perform better reciprocal anchorage and thus higher movement predictability.

The use of attachment significantly increases aligner retention. Conventional rectangular attachments on premolars have the best effect in terms of keeping the aligners on the teeth and retaining the aligner significantly stronger than similar attachments on the incisors.

One should not forget that the efficacy of orthodontic tooth movement is influenced by multiple other factors, including aligner thickness [43], aligner treatment staging, timing and the amount of interproximal enamel reduction, patient cooperation and the use of other auxiliaries (elastics, temporary anchorage devices (TADs) and segmental orthodontic appliances).

The other important issue to discuss is patient acceptance of a large number of composite attachments and their visibility. Too many attachments or too large a size on the buccal surface of the teeth can compromise the invisibility of the aligner treatment [44]. As research shows, this is one of the most important features taken into account by the patient when choosing an aligner treatment [45]. Therefore, this must be considered in treatment design.

A systematic review on the effect of orthodontic tooth movement was published at the beginning of 2022 [46] based on a search conducted in 2020, where only five studies were included, as only clinical trials were considered. In the present systematic review, in addition to new clinical publications, FEM studies are also included. Already in 2017, Goto et al. [47] indicated that FEM should be the target method for evaluating the effect of attachments on tooth movement. In the eyes of the authors of the present study, FEM studies are crucial in properly developing public domain knowledge in the field of aligners. First, all moderate and complex aligner treatments are based on virtual set-up and staging, which is intended to fully reflect what would happen in the oral cavity of a patient. FEM studies also allow the introduction of variables that can cause changes in orthodontic tooth movement, which makes them much more reliable than a standard virtual set-up. Moreover, current clinical studies regarding aligners are related to increased costs. On the other side, to stay objective, studies should not be co-financed by the manufacturers of aligners. That means that a patient is most often paying more than in the case of standard fixed appliances (about 3500EUR for aligner treatment within the European Union) [48], obviously making patients less likely to participate in research. However, the results of FEM studies can only be taken as a general description of the phenomena they describe and not as absolute values. While the selection of a patient with a specific malocclusion is correct, variables such as the thickness of the aligner or the dimensions of the attachments are so diverse between the studies that the results should be viewed as a relative reaction of the tissues and not an exact calculation.

The impact of risk of bias and publication bias on the interpretation of the result leads to the finding that due to the method of the studied phenomenon (the presence and shape of a composite attachment), it is difficult to conduct prospective high-quality clinical trials. It would be extremely difficult to obtain a suitable control group that agreed to possibly worse treatment results without the use of attachments or the use of presumably less efficient shapes. In light of the current scientific evidence in the form of highly probable computer simulations, conducting such a study should not be approved by the bioethics committee. Due to the fact that many of the FEMs, in vitro studies and retrospective studies meet the evidence quality requirements provided in the scales for a given type of study, the results contained in them should be considered as likely to occur during aligner treatment.

### 4.3. Limitations and Direction of the Future Treatment

Possible limitations of the present study may be due to the fact that the published clinical studies are based on data collected from different periods of time and are thus mixing together treatments performed according to different aligner protocols or are based on old protocols—similar to the study published in 2019 based on the Invisalign G6 protocol. By the time of writing this manuscript, the current Invisalign treatment protocol is 8/8+, and scientific evidence concerning the improvement between the protocols is purely descriptive [49], without any clinical trials that assess the effect of specific types of movements. The authors of the study cited indicated that overcorrection should still be used. The impact of currently used aligner protocols on the treatment results has no scientific evidence yet.

There are case series published in the orthodontic literature pointing to the advantages of hybrid treatments, including the use of aligners with TAD-supported anchorage devices, tubes or segmental orthodontic appliances (3–4 brackets) [50,51,52]. In a recently published handbook for aligner treatment, such a procedure is often even advised [53]. However, no studies provide a scientific background for aligner-only or hybrid treatments.

## 5. Conclusions

1. The presence of attachments significantly improves aligner retention and orthodontic tooth movement

2. Conventional attachments, with at least one beveled edge, are beneficial both for tooth movement and for anchorage. On anterior teeth, it is worth placing attachments on the lingual surfaces, both for therapeutic and aesthetic indications.

3. It seems reasonable to recommend liquid composites or orthodontic bonding composites with as much filler as possible bonded with a rigid transfer tray for better accuracy and composite bond strength.

4. Future studies could concentrate on clinical tooth movement assessment in aligner therapy, the efficacy of new generations of aligner treatment protocols and eventual indications for “hybrid treatment”.

## Figures and Tables

**Figure 1 ijerph-20-04481-f001:**
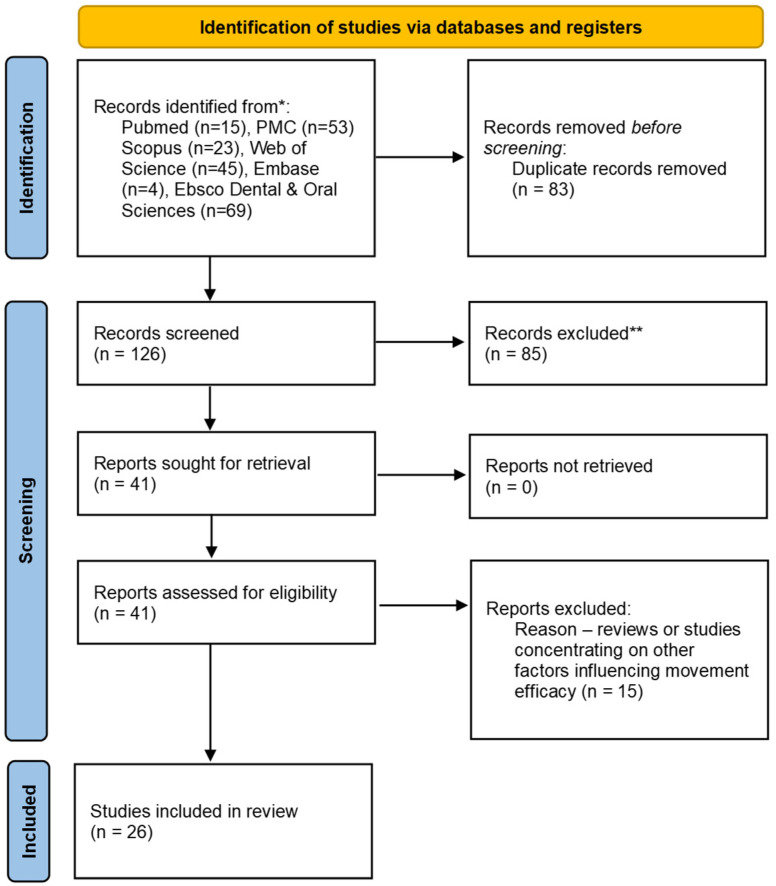
Prisma 2020 Flow Chart. *—detailed search strings presented in Chapter 2.1. **—studies not related to the review topic at all.

**Table 2 ijerph-20-04481-t002:** Included studies regarding influence of composite attachment presence on movement efficacy.

Authors and Year	Type of the Study	Studied Phenomena	Subjects within the Study	Results and Clinically Relevant Conclusions
Ahmed et al. 2022 [21]	Finite element analysis	The influence of attachment positioning on movement efficiency (incisors intrusion/retrusion) and stress distribution	Model without attachments Model with labial attachments Model with palatal attachments Model with labial and palatal attachments	The most effective retraction was obtained by using palatal attachments. In the labial attachment model, the stress was concentrated in the middle third, while in all other models, this occurred in the cervical third. The model without an attachment generated the highest stresses. Placing the attachments on the labial surface can help avoid uncontrolled tipping.
Ayidağa et al. 2021 [22]	Finite element analysis	The influence of attachment shape on movement efficiency (molar distalization)	Models with no composite attachment Vertical rectangular attachment positioned on the buccal surface of the maxillary 1st molar Guideline attachment positioned on the buccal surface of the first maxillary molar.	All of the models were used to determine the effect on the periodontal ligament and the bone. The “no attachment” model was characterized by the lowest amount of desired translation on the *y*-axis and by the highest undesired distal tipping movement. In the rectangular attachment model, the tooth moved significantly more along the y-axis, with a smaller component of tipping and distal rotation. In the guided attachment model, the amount of movement along the *y*-axis was similar; however, there was the smallest range of undesired movements in the form of tipping or rotation. PDL was the point stress concentration in the first 2 groups, while in the guideline attachment group, the stress was equally distributed.
Cortona et al. 2020 [23]	Finite element analysis	The influence of attachment positioning on movement efficiency (molar rotation)	Without attachments Single vertical 3 mm attachment was placed on the buccal surface of element 45 Three vertical 3 mm attachments were placed on the buccal surfaces of teeth 44 to 46. In all three models, the activation of 1.2 degrees and 3 degrees per aligner was considered. In all models, an attempt was made to derotate the second premolar in the fourth quadrant.	The model with a single attachment on 45 and 1.2 degrees of aligner activation was the most efficient, followed by the three-attachment model (at the same degree of activation).Aligner activation should not exceed 1.2 degrees to achieve good control of movement and reasonable stress in periodontal structures.
Costa et al. 2020 [24]	In vitro experimental study	The influence of attachment shape and positioning on movement efficiency (incisor extrusion)	Three models were designed to evaluate which attachment design enables the most effective upper incisor extrusion Rectangle with 8 mm² on its gingival face and a 3 mm thickness from the dental surface to the frontal face A 2 × 4 × 1 mm^3^ cuboid, associated with two 0.87 × 4 mm^2^ rectangular planes angled at 45° with the cuboid surface A frontal face without edges and less protrusive, with a vestibular length of 3.32 mm	Different attachment geometries generate forces with significantly different intensities and directions. The third attachment had the best mechanical performance among the three models evaluated for extrusion.
Dai et al. 2019 [25]	Randomized controlled clinical trial	The influence of attachment shape and positioning on movement efficiency (premolar extraction space closure)	G6-optimized attachment on the first molar A 3 mm vertical rectangular attachment on the first molar A 3 mm horizontal attachment on the first molar A 5 mm horizontal attachment on the first molar	G6-optimized attachments, together with horizontal attachments, showed similar efficacy and control in molar angulation. The vertical attachment had the biggest difference between the planned and achieved tooth movement and showed the worst anchorage control (the highest degree of molar tipping).
Dasy et al. 2015 [26]	In vitro experimental study	The influence of attachment shape on aligner retention	Ellipsoid attachments (height: 3 mm, width: 2 mm, depth: 1 mm) Rectangular attachments, which were beveled toward the incisal edge (height: 2 mm, width: 3 mm, depth: 0.25 mm incisally and 1.25 mm gingivally). No attachments	The use of beveled attachments significantly increased retention. Ellipsoid attachments showed no significant effect on aligner retention with regard to an aligner with no attachments.
Fan et al. 2022 [27]	Finite element analysis	The influence of attachment positioning on movement efficiency (molar intrusion)	No attachment Buccal attachment Palatal attachment, Buccal and palatal attachments	The presence of an attachment is essential for clear aligners to intrude on the molars. Combined buccal and palatal attachments could effectively prevent buccal and palatal tipping and showed the best efficiency in terms of intruding on the molars. The second molar showed an unavoidable tendency to tip mesially, regardless of the attachment position. Thus, double attachment is advisable.
Ferlias et al. 2022 [28]	In vitro experimental study	The influence of attachment shape and positioning on movement efficiency (premolar rotation)	A total of 11 different types of attachment and one with no attachment Invisalign^®^ “Bevelled” (Bevelled, 3.5 × 1.5 × 1 mm), “Horizontal Ellipsoid” (HEllipsoid, 3 × 2 × 1 mm), “Vertical Ellipsoid” (VEllipsoid 3 × 2 × 1 mm), “Elliptical Pair” (ElliPair, 2 × 2 × 1 mm/each), “Hemi-elliptical Right” (HemiEllipR, 2 × 2 × 1 mm), “Hemi-elliptical Left” (HemiEllipL, 2 × 2 × 1 mm), “Horizontal Rectangular Left” (HRecL, 3.5 × 1.5 × 1 mm), “Horizontal Rectangular Right” (HRecR, 3.5 × 1.5 × 1 mm), “Vertical Rectangular Down” (VRecDOWN, 3.5 × 1.5 × 1 mm), “Vertical Rectangular Up” (VRecUp, 3.5 × 1.5 × 1 mm). “3Shape^®^ Box” (3Shape, 3.5 × 1.5 × 1 mm) “No Attachment”	The rotations above 1° generate moments that are too high from a clinical point of view. Aligner steps of no more than 1–1.5° should be recommended for effective derotation of a premolar. The vertical rectangular attachments perform best when derotating a premolar due to their large flat active surface but receive the most side effects in terms of tipping, torque and intrusive force. Derotation of a premolar without any attachment was less efficient despite showing the least side effects. When a premolar was mesially rotated, the attachment producing the highest intrusive force was 3Shape. When the tooth was rotated distally, most attachments again received an intrusive force, while the 3Shape attachment displayed an extrusive force. When the tooth was rotated mesially, all setups received a buccal root torque, with the highest seen with the vertical rectangular attachment and the smallest with the beveled one. In the other direction, for the distally rotated tooth, a moment of lingual root torque was observed in all setups.
Garino et al. 2016 [29]	Randomized clinical trial	The influence of attachment shape and positioning on movement efficiency (molar movement along a different axis)	A total of 30 non-growing patients in need of distalization after third-molar extraction. Rectangular composite attachments were placed on all distalized teeth from canine to second molar (five attachments per quadrant) Rectangular attachments were used only on the first and second premolars and the first molar (three attachments per quadrant)	Although the groups did not differ in the amount of distalization, minimizing distal crown tipping and preventing molar extrusion, minimizing anterior anchorage loss and reducing undesirable changes in lower facial height were more effective in patients with all five teeth bonded with attachments. This approach seems promising in patients in need of 2–3 mm of distalization.
Gomez et al. 2015 [30]	Finite element analysis	The influence of attachment presence on stress distribution (canine distalization)	No attachment model Two optimized ellipsoid optimized attachment models from a random Invisalign case	The displacement of the model with attachments was equivalent to typical distal bodily movement. The displacement of the model without attachments was equivalent to typical uncontrolled distal crown tipping, with almost no reaction of the root.
Karras et al. 2021 [31]	Retrospective cross-sectional study	The influence of attachment type (optimized vs. conventional) on movement efficacy	Included a total of 120 teeth arches, from which 163 teeth qualified for optimized rotation attachments (43%): 72 conventional rotation attachments (19%), 81 optimized extrusion attachments (21%) and 66 conventional extrusion attachments (17%)	For all tooth movements and attachment types, the mean predicted values were significantly larger than the mean achieved values. The least accurate tooth movement was mandibular canine extrusion with a conventional attachment (16.1%). The most accurate tooth movement was the extrusion of the maxillary central incisor with a conventional attachment (73.9%), followed closely by the rotation of the maxillary premolar with an optimized attachment (72.8%). Overall, the optimized attachments enabled achieving better results in terms of rotation movement, while conventional attachments performed better in extrusion movements.
Kim et al. 2020 [32]	Finite element analysis	The influence of attachment shape and positioning on movement efficiency and stress distribution (lower canine rotation)	A virtual model with two attachments on the lower canine. The shape of the attachment for rotation had a plane perpendicular to the direction of rotation with the surfaces to which the load was applied. The models were classified as having angles of 90 degrees, 65 degrees and 45 degrees to the attachment surface of the teeth. There were four types of attachments for the torque tested; four shapes were half round, half round at the cross-section, half round at the cross-and longitudinal sections, and a lower bevel of 45°.	A desirable stress distribution was observed when there was a high contact area between the attachment and the aligner. Torque control and intended movement were achieved when the attachments were positioned on the lingual surface rather than on the buccal surface of the canines; thus the attachment used in the aligner treatment of the rotated canine is a cylinder form bonded to the lingual surface of the canine. In intrusion, a better control of the movement is achieved by bonding attachments to both the buccal and lingual surfaces of the canine.
Kravitz et al. 2008 [33]	Randomized clinical trial	The influence of attachment shape and positioning on movement efficiency (canine rotation)	No attachments Interproximal reduction without attachments Canines with attachments without IPR	Vertical, ellipsoid attachments and interproximal reduction do not significantly improve the accuracy of canine rotation with the Invisalign system.
Laohachaiaroon et al. 2022 [34]	Finite element analysis	The influence of attachments shape on movement efficiency and stress distribution (upper central incisors extrusion)	Without any composite attachment Rectangular beveled attachment on the labial surface of the central and lateral incisors Ellipsoid attachment on the labial surface of the central and lateral incisors Horizontal rectangular attachment on the labial surface of the central and lateral incisors	When considering the incisal edge as a reference, the model with the horizontal rectangular attachment had the greatest extrusive movement (0.037991 mm) followed by the model with the ellipsoid attachment (0.037606 mm) and the model with the rectangular beveled attachment (0.036786 mm). The model without a composite attachment demonstrated little intrusive movement (0.000105 mm). The differences were very small and were not clinically significant. The stress patterns were also similar in all three attachment models.
Rossini et al. 2020 [35]	Finite element analysis	The influence of attachment use on movement efficiency and stress distribution (maxillary second molar distalization)	No attachments Vertical 3 mm vertical attachments from the canine to the first molar Vertical 3 mm vertical attachments from the canine to the second molar	The attachments are mandatory to control the bodily movement of a second molar. Attachments should be used to reinforce the anchorage units and to function as active units on sequential distalizing molars. The configuration of attachments in the whole segment, from the canine to the 2nd molar, represents the most promising model for Class II correction via maxillary molar distalization.
Rossini et al. 2021 [36]	Finite element analysis	The influence of attachment presence on movement efficiency and stress distribution (incisors extrusion)	Without attachments Horizontal rectangular attachments only on incisors Rectangular attachments from the second molar to the canine Rectangular attachments from the second molar to the canine used together with optimized extrusion attachments on incisors Rectangular attachments from the second molar to the canine used together with rectangular buccal horizontal attachments on the incisors Rectangular attachments from the second molar to the canine used together with rectangular palatal horizontal attachments on the incisors	The presence of 3 mm rectangular horizontal attachments on the buccal or palatal surface of the upper incisors with additional rectangular vertical attachments in the lateral from the canine to the second molar seemed to produce the most efficient force system to extrude incisors with minimal aligner deformation. Standard attachments seem to be more accurate than optimized ones. The most efficient configurations showed the need for the use of attachments on posterior teeth in order to obtain better anchorage.
Savignano et al. 2019 [37]	Finite element analysis	The influence of attachments on movement efficiency and stress distribution during upper incisors extrusion	Without attachments Horizontal rectangular palatal attachment Horizontal rectangular buccal attachment Ellipsoid buccal attachment	The extrusion of an upper central incisor cannot be achieved without any attachment. There was no clear difference between the rectangular and ellipsoid attachments. The position of the attachment showed a stronger influence on the outcome compared to the shape (palatal instead of buccal).
Simon et al. 2014 [38]	Retrospective cross-sectional study	The influence of attachment shape and positioning on movement efficiency (incisors torque, lateral teeth derotation and distalization)	Patients treated with the use of an attachment Patients with no auxiliaries were used (except incisor torque, in which Power Ridges were applied)	The use of attachments can significantly improve the efficiency of planned tooth movement. However, clinically planned movements are rarely completed. The crucial factors that influence the efficiency of aligner therapy are patient compliance and the reasonable, split staging of planned moves <1.5° of rotation per aligner, 1° aligner for incisor torque and up to 0.25 mm/aligner for distalization. Invisalign treatment usually needs refinements to achieve planned positions of the teeth.
Smith et al. 2022 [39]	Retrospective cross-sectional study	The influence of attachments on tooth movement efficiency (lower incisor tipping)	66 lower incisors in 42 non-extraction aligner patients A total of 21 incisors with vertical attachments A total of 45 incisors without any attachments	It is possible to move roots using Invisalign^®^ but not as predictably as ClinCheck^®^ suggests. Moreover, the average amount of root movement achieved was substantially less than predicted. Vertical rectangular attachments are recommended when a large range of root movement is planned. Attachments improve the possibility of translating the root apex.
Takara et al. 2022 [40]	In vitro experimental study	The influence of attachment shape on aligner retention	A total of 22 different models with 11 different types of attachments were placed on the lateral incisors or the first premolars and included a model with no attachments	Attachments significantly increase aligner retention. The easiest way to remove an aligner is from the lingual side of the first molar. It is difficult to remove the aligner by trying to lift it in the area of the incisors. However, the attachment on the lateral incisor may not contribute to the gripping force of the aligner when it is removed by lifting on the labial surface of the upper first molar. The retention of the aligner is influenced by the height, width and angulation of the attachment. The retention is superior when the bevel angle is close to the right angle.
Yokoi et al. 2019 [41]	Finite element analysis	The influence of attachment shape and positioning on movement efficiency (diastema closure)	No attachment Double contrary attachments on the labial surface of the central incisor	The use of attachments limited unplanned root movement and tooth tipping, increasing the effectiveness of diastema closure. In the aligner used with the attachments, the incisor overlapped completely on the target position in FEM, meaning that the efficacy of movement was almost 100%. However, the attachment did not influence the initial movement of the tooth, showing significant differences with an ongoing time simulation.

**Table 3 ijerph-20-04481-t003:** Quality assessment according to MSQFE assessment tool for finite element analysis.

	Question	Ahmed et al. 2022 [21]	Ayidağa et al. 2021 [22]	Cortona et al. 2020 [23]	Fan et al. 2022 [27]	Gomez et al. 2015 [30]	Kim et al. 2020 [32]	Laohachaia-Roon et al. 2022 [34]	Rossini et al. 2020 [35]	Rossini et al. 2021 [36]	Savigniano et al. 2019 [37]	Yokoi et al. 2019 [41]
**Study Design and Presentation of Findings**												
1	Was the hypothesis/aim/objective of the study clearly described?	Yes	Yes	Yes	Yes	Yes	Yes	Yes	Yes	Yes	Yes	Yes
2	Were all analyses planned at the outset of the study? Answer NO for unplanned analysis/sub-analysis, unable to determine.	Yes	Yes	Yes	Yes	Yes	Yes	Yes	Yes	Yes	Yes	Yes
3	If data dredging (establish objectives, hypothesis and endpoint parameters without scientific reason) was used, was the spectrum of the data justified by any concepts? Answer YES if no data dredging, NO if unable to determine	Yes	Yes	Yes	Yes	Yes	Yes	Yes	Yes	Yes	Yes	Yes
4	Were all the outcome measures and parameters (including all data reduction methods or derived parameters) clearly described and defined in the Objectives or Methods section? Answer NO if they are only defined in results or discussion	Yes	Yes	Yes	Yes	No	Yes	Yes	Yes	Yes	Yes	Yes
5	Were the time points or period for all the outcome measures clearly described? Answer YES if not applicable	No	No	Yes	Yes	Yes	No	Yes	Yes	Yes	Yes	Yes
6	Were the main outcome measures appropriate to describe the targeted conditions? Answer NO if unable to determine	Yes	Yes	Yes	Yes	Yes	Yes	Yes	Yes	Yes	Yes	Yes
7	Were the key findings described clearly?	Yes	Yes	Yes	Yes	Yes	Yes	Yes	Yes	Yes	Yes	Yes
8	Were all the contour plots that were used for comparison presented with the same color scale?	Yes	Yes	Yes	Yes	Yes	Yes	Yes	Yes	Yes	Yes	Yes
**Subject Recruitment**												
9	Were the characteristics of the model subject clearly described?	Yes	Yes	Yes	Yes	Yes	Yes	Yes	Yes	Yes	Yes	Yes
10	Were the principal confounders of the model subject clearly described?	Yes	Yes	Yes	Yes	Yes	Yes	Yes	Yes	Yes	Yes	Yes
11	Was the model subject participated in the study representative of the population with the targeted clinical conditions or demographic features? (e.g., answer NO if simulating a pathology by modifying a normal subject model; or scaling an adult model to a child model)	Yes	Yes	No	Yes	Yes	Yes	Yes	Yes	Yes	Yes	Yes
12	Were the targeted intervention or clinical condition clearly described?	Yes	Yes	No	Yes	Yes	Yes	Yes	Yes	Yes	Yes	Yes
**Model Reconstruction and Configuration**												
13	Was the model reconstruction modality for the body parts and all other items, such as implants, clearly described (e.g., MRI, 3D-scanning, CAD)?	Yes	Yes	No	Yes	No	Yes	Yes	Yes	Yes	Yes	Yes
14	Were all important technical specifications (e.g., resolution) for the reconstruction modality clearly described?	Yes	Yes	Yes/No—material described clearly, but not mandibular model	No	No	Yes	Yes	Yes	Yes	Yes	Yes
15	Was the posture or position of the body parts controlled during the acquisition process (e.g., MRI, CT) of the model reconstruction?	Yes	Yes	No	No	No	Yes	No	Yes	No	Yes	No
16	Were the model reconstruction methods for all components clearly described including those requiring additional procedures (e.g., connecting points for drawing ligaments from MRI)?	No	No	Yes/No	Yes	Yes	Yes	No	Yes	No	Yes	No
17	Were the orientation or relative position among the components of the model assembly (where appropriate) clearly described? Answer YES if not applicable	Yes	Yes	Yes	Yes	Yes	Yes	No	Yes	No	Yes	No
18	Was the type of mesh for all components, including the order of magnitude of the elements, clearly described?	No	Yes	No	Yes	Yes	Yes	Yes/No	No	No	Yes	Yes
19	Were the material properties for all components clearly described and justified? (e.g., with reference)	Yes	Yes	No	Yes	Yes	Yes	Yes	No	No	Yes	Yes
20	Were all the contact or interaction behaviours in the model clearly described and justified?	Yes/No	Yes	Yes/No	Yes	Yes	Yes	Yes	Yes	Yes	Yes	Yes
**Boundary and Loading Condition (Simulation)**												
21	Were the boundary and loading conditions clearly described?	Yes	Yes	Yes	Yes	Yes	Yes	No	No	No	Yes	Yes
22	Was the boundary and loading condition sufficiently simulating the common activity/scenario of the conditions? (e.g., if the research or inference is targeted to ambulation or daily activities, simulations of balanced standing or pre-set compressive load are insufficient)	Yes	Yes	Yes	No	Yes	Yes	No	No	No	Yes	Yes
23	Was the model driven by the boundary condition acquired from the same model subject?	Yes	Yes	Yes	Yes	Yes	Yes	No	No	No	Yes	Yes
24	Was loading condition on the scenario sufficiently and appropriately considered in the simulation? (e.g., muscle force, boundary force, inertia force)	Yes/No	Yes/No	Yes	Yes	Yes	Yes	No	Yes	Yes	Yes	Yes
25	Was the loading condition acquired from the same model subject?	Yes	Yes	Yes	Yes	Yes	Yes	No	Yes	Yes	Yes	Yes
26	Was the software (e.g., Abaqus, Ansys), type of analysis (e.g., quasi-static, dynamic), and solver (e.g., standard, explicit) clearly described? (Solver can be regarded as clearly described if it is obvious to the type of analysis)	Yes	Yes	Yes	Yes	Yes	Yes	No	Yes	Yes	Yes	Yes
**Model Verification and Validation**												
27	Were the methods of mesh convergence or other verification tests conducted and clearly described?	Yes	Yes	Yes	No	Yes	No	No	Yes	Yes	Yes	No
28	Were the model verification conducted and results presented clearly, and that the model was justified acceptable?	Yes	Yes	No	No—just mentioned carrying them out	Yes	Yes	No	Yes	No—just mentioned carrying them out	Yes	No
29	Was direct model validation (with experiment) conducted and described clearly? Answer YES if the authors had direct model validation previously with reference.	Yes	Yes	Yes	No	Yes	No	No	Yes	Yes	Yes	No
30	Were the model validation conducted and results presented clearly, and that the model was justified acceptable?	Yes	Yes	No	Yes/No	Yes	No	No	Yes	Yes	Yes	Yes
31	Were the model prediction or validation findings compared to relevant studies?	Yes	Yes	Yes	No	Yes	No	No	Yes	Yes	Yes	Yes
**Model Assumption and Validity**												
32	Were the model assumptions or simplifications on model reconstruction/configuration and material properties discussed?	Yes	Yes	No	Yes	Yes	Yes	Yes	Yes	No	Yes	No
33	Were the model assumptions or simplifications on the boundary and loading conditions discussed?	Yes	Yes	Yes/No	Yes	Yes	Yes	Yes	Yes	No	Yes	No
34	Were the limitations of model validation discussed? (e.g., differences in case scenario; differences between validation metric and primary outcome)	No	Yes	No	Yes/No	Yes	No	No	Yes	No	Yes	Yes
35	Was the limitation on external validity, single-subject, and subject-specific design discussed?	Yes	Yes	No	No	Yes	Yes	No	Yes	No	No	Yes
36	Were there any attempts to improve or discuss internal validity (such as the mesh convergence test), uncertainty and variability in the study?	Yes	Yes	Yes	No	Yes	No	No	Yes	Yes	Yes	No
37	Was there any discussion, highlights or content on the implications or translation potential of the research findings? Answer NO if there are only bold claims without making use of the result findings or key concepts	Yes	Yes	Yes	Yes	Yes	Yes	No	Yes	Yes	Yes	Yes
Sum:		33.5	34.5	24.5	26.5	33	30	18.5	32	24	36	28

**Table 4 ijerph-20-04481-t004:** Quality assessment of in vitro studies according to QUIN assessment tool.

Criteria No.	Criteria	Alsaud et al. 2022 [16]	Chen et al. 2021 [18]	Costa et al. 2020 [24]	D’Antò et al. 2019 [19]	Dasy et al. 2015 [26]	Ferlias et al. 2022 [28]	Takara et al. 2022 [40]	Valeri et al. 2022 [20]
1	Clearly stated aims/objectives	2	2	2	2	2	2	2	2
2	Detailed explanation of sample size calculation	2	2	0	2	0	0	0	0
3	Detailed explanation of sampling technique	2	2	2	2	1	1	2	2
4	Details of comparison group	2	2	1	2	2	2	2	2
5	Detailed explanation of methodology	2	2	1	2	2	2	2	2
6	Operator details	0	1	0	2	0	2	2	2
7	Randomization	0	1	0	0	0	0	0	0
8	Method of measurement of outcome	2	2	2	2	2	2	2	2
9	Outcome assessor details	2	2	2	2	1	1	2	0
10	Blinding	0	0	0	0	0	0	0	0
11	Statistical analysis	2	2	2	2	2	2	2	2
12	Presentation of results	2	2	2	2	1	2	2	2

**Table 5 ijerph-20-04481-t005:** Evaluation of included studies according to Cochrane Risk of Bias Tool for Randomized Controlled Trial.

	Bruno et al. 2021 [21]	Dai et al. 2019 [25]	Garino et al. 2016 [29]	Kravitz et al. 2008 [33]
Random sequence generation	LOW	SOME CONCERNS	LOW	LOW
Allocation concealment	LOW	HIGH	LOW	SOME CONCERNS
Blinding of participants and personnel	HIGH	HIGH	HIGH	HIGH
Blinding of outcome assessment	SOME CONCERNS	HIGH	SOME CONCERNS	HIGH
Incomplete outcome data	LOW	LOW	LOW	LOW
Selective reporting	LOW	LOW	LOW	HIGH
Other bias (why)	The bias comes from the difference between the bonding procedure and analyzing the data, as the templates and attachments differ from each other	Great loss to follow-up	The bias comes from the difference between the bonding procedure and analyzing data, as attachments are noticeable in the mouth	The pretreatment model was obtained by the indirect and direct methods, resulting in an uneven level of accuracy.
Risk of bias judgement	SOME CONCERNS	HIGH	SOME CONCERNS	HIGH

**Table 6 ijerph-20-04481-t006:** Newcastle–Ottawa Quality Assessment Form for Cross-Sectional studies.

Study		Karras et al. 2021 [31]	Simon et al. 2014 [38]	Smith et al. 2022 [39]
Selection	Representativeness of the cases	*	*	*
	Justified sample size	*	0—not justified	*
	Non-respondents	*	*	*
	Ascertainment of the exposure (risk factor)	**	*	**
Comparability	The subjects in different outcome groups are comparable based on the study design or analysis. The confounding factors are controlled	**	**	**
		Yes, the group was homogenous regarding the studying phenomena	Yes, the group was homogenous regarding the studied phenomena	Yes, the group was homogenous regarding the studied phenomena
Outcome	Assessment of the outcome	**	**	**
	Statistical test	*	*	*
Total		10	8	10

* one point, partial compliance with the criterion; ** two points, full compliance with the criterion.

**Table 7 ijerph-20-04481-t007:** Clinical guide for effective composite attachment placement basing on information in included studies.

Tooth	Movement	Attachment Type and Position	Example of Attachment in Orthodontic CAD Software [42]
Upper incisors	Retrusion/intrusion/extrusion	Conventional, rectangular, beveled, horizontal attachments, preferably placed on the palatal surface and preferably with auxiliary attachments on the posterior teeth	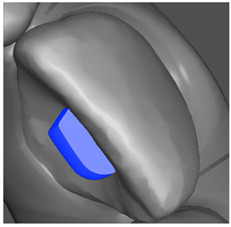 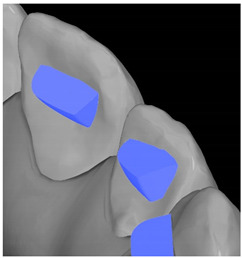
Upper incisors	Diastema closure	Double contrary, sassy attachments on the labial surfaces of the central incisor	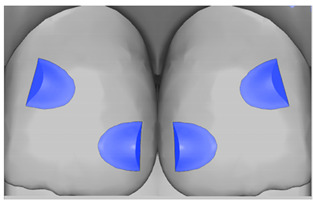
Lower incisors	Intrusion/extrusion	Conventional, rectangular, beveled, horizontal attachments, preferably placed on the palatal surface and preferably with auxiliary attachments on the posterior teeth	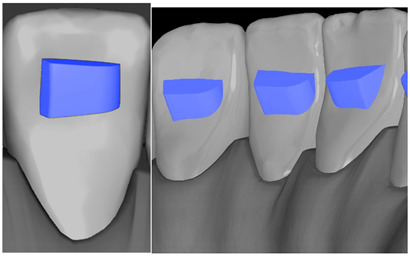
Lower incisors	Tipping	Conventional, rectangular, vertical attachments, preferably placed on the palatal surface	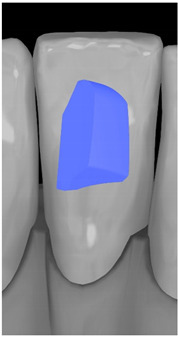
Canine	Intrusion/Extrusion	Optimized, one-side-beveled attachment	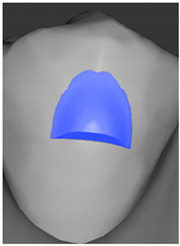
Canine	Distalization	Two ellipsoid sassy optimized attachment or beveled angled single optimized attachment, preferably placed together with conventional vertical attachments on adjacent teeth	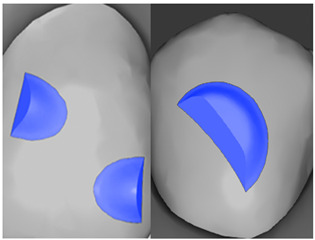
Canine	Rotation	Conventional ellipsoid one-side-beveled attachment on lingual surface of the tooth/Conventional vertical beveled attachments on the lingual and buccal surfaces of the tooth	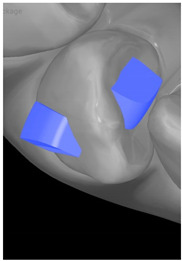
Premolar	Rotation	Vertical rectangular attachments/optimized vertical attachments	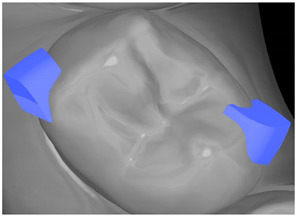 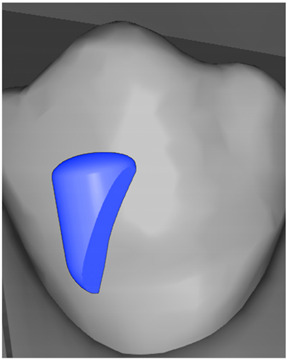
Molar	Intrusion/extrusion	Two conventional vertical attachments—one on the palatal surface, one on the buccal surface/optimized vertical attachments	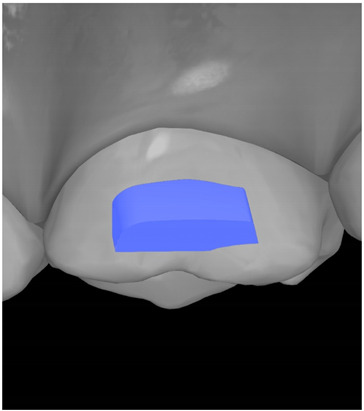 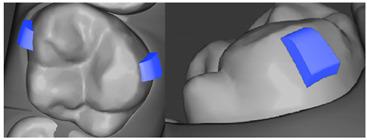 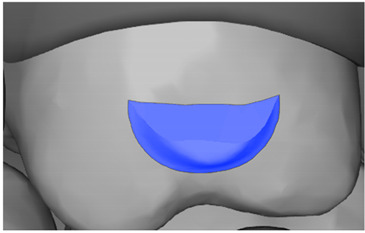
Molar	Mesialization	Optimized horizontal attachment	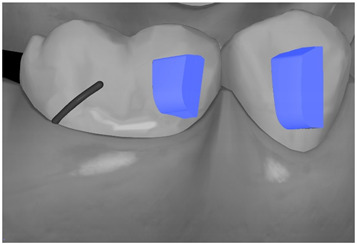
Molar	Rotation	Three vertical 3 mm attachments placed on the buccal surfaces of the teeth to be rotated and the adjacent teeth	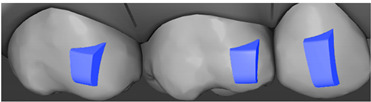
Molar	Distalization	Preferably, conventional 3 mm vertical or optimized attachment on the second molar placed together with conventional (preferably vertical) attachments from the canine to the second molar	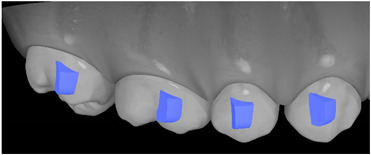 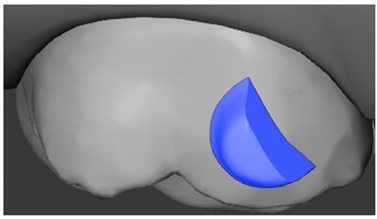
Molar	Distalization+ Distorotation	Vertical 3 mm attachment placed on the buccal surface	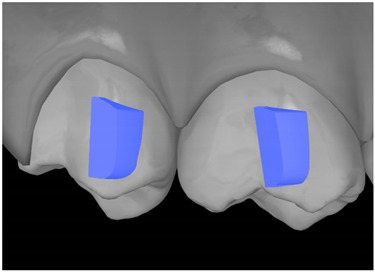
Molar	Uprighting	Horizontal 3 mm attachment located on the buccal surface at an angle of 45°	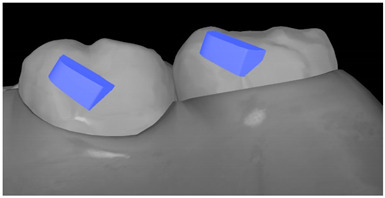

## Data Availability

Data from the present study can be provided upon reasonable request.

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
