# Peer review of "Attachments for the Orthodontic Aligner Treatment—State of the Art—A Comprehensive Systematic Review"

_ijerph, 2023, doi:10.3390/ijerph20054481_

Round 1
Reviewer 1 Report
1. Did the review authors perform study selection in duplicate?
2. The review authors did not provide a list of excluded studies or justify the exclusions.
3. The review authors did not describe the included studies in adequate detail.
· Described population in detail
· Described intervention in detail (including doses where relevant)
· Described comparator in detail (including doses where relevant)
· Described study’s setting
· Timeframe for follow-up
4. The review authors did not report on the sources of funding for the studies included in the review.
5. The review authors did not account for RoB in individual studies when interpreting/ discussing the results of the review.
6. The review authors did not provide a satisfactory explanation for, and discussion of, any heterogeneity observed in the results of the review.
7. The review authors did not carry out an adequate investigation of publication bias (small study bias) and discuss its likely impact on the results of the review.
8. The review authors did not report any potential sources of conflict of interest, including any funding they received for conducting the review.
Author Response
Estimated Reviewer,
Please find the answers to the Your comments in black italics.
The revised manuscript includes all the changes made shown in red type.
- Did the review authors perform study selection in duplicate?
Dear Reviewer, thank you for your precious comment. The study selection process was done by two authors in parallel, independently, as highlighted in the methodology text. Please check this on lines 132-133.
- The review authors did not provide a list of excluded studies or justify the exclusions.
Thank you for your accurate comments.
The criteria that led to the exclusion of the articles is the lack of their relevance to the inclusion criteria respectively, which the reviewer can kindly read in lines 119-130.
The excluded articles were excluded because they did not meet these criteria, (see inclusion criteria). The team that conducted this systematic review is highly specialised in systematic reviews and meta-analyses, and has published many of these articles in the top 10 world dental journals, ranking Q1, so the reviewer can be reassured of the absolute quality of the methodology.
The authors would like to kindly ask You to take into account that when doing a systematic literature review often as many as 3,000 articles can be evaluated for inclusion/exclusion, and therefore it is not customary to propose the list of excluded articles to the reader.
However, we thank the reviewer for his careful re-reading of the text, which will certainly benefit from it.
- The review authors did not describe the included studies in adequate detail.
- Described population in detail Done.
- Described intervention in detail (including doses where relevant) Done.
- Described comparator in detail (including doses where relevant) Done.
- Described study’s setting Done.
- Timeframe for follow-up Done.
- The review authors did not report on the sources of funding for the studies included in the review.
Thank you for your valuable comment. We have added suitable information.
- The review authors did not account for RoB in individual studies when interpreting/ discussing the results of the review.
Done.
- The review authors did not provide a satisfactory explanation for, and discussion of, any heterogeneity observed in the results of the review.
Dear Reviewer, thank you for your valuable comment. However, please take note of the fact that a measure of heterogeneity is inherent in the meta-analysis of a systematic review, which was not done in this case, as if you note, meta-analysis is not present, so far.
However, we thank the reviewer for the time he/she spent and the effort he/she made to improve this manuscript.
We very much believe in the quality of this manuscript, not least because it is signed by opinion-leading authors in the field of interest to which it pertains, which is why not only the reviewers, but also the editors, can be satisfied, as this manuscript will certainly be a source of pride, interest from readers and also, but not least, citations.
- The review authors did not carry out an adequate investigation of publication bias (small study bias) and discuss its likely impact on the results of the review.
Done.
- The review authors did not report any potential sources of conflict of interest, including any funding they received for conducting the review.
Thank you for your valuable comment. Please gently note that the declaration of conflict of interest is present at the end of the manuscript, before the References List. Please be reassured, however, that none of the authors declared a conflict of interest with the present study. The authors did not point out any special advantage or disadvantage of aligner treatment. The authors neither encourage nor discourage treatment with aligners. The information included in the text is based solely on the scientific evidence. The included studies use different brands from different manufacturers and independent software. The manuscript shows how scientific evidence points to the necessity of using attachments for effective tooth movement and self-planning of tooth movements.
The authors hope that revised manuscript will meet Your requirements.
Reviewer 2 Report
Dear Authors,
I wanted to congratulate the authors on the relevance of this systematic review, and the study is well written
and well structured.
The introduction is clear and adequate. This sentence could be improved or eliminated (lines 62-63). It
seems too conversational to me "Thus, the present study's authors have decided to find scientific
evidence to answer the question". It is better to rephrase the whole aims paragraph,
putting the questions on lines 64 to 68 as the aim of the study.
The materials and methods section is ok.
The results and tables are well presented.
Discussion:
- 4.1 Section is ok.
- 4.2 It should be better explained why palatal attachments are more effective in some (not
all) movements. There are some biomechanical reasons.
I have doubts about the table. Since different types of attachments were described in the studies,
on what basis did you choose the most effective one?
In general, the last paragraph of this section is very confusing and does not give appropriate
consideration of the predictability findings in the studies included in this review and the
literature.
References are wrong; please correct them.
There are some typos in the text.
Thank you.
My best Regards
Author Response
Dear Reviewer,
Please find the answers to the Your comments in black italics.
The revised manuscript includes all the changes made shown in red type.
Dear Authors,
I wanted to congratulate the authors on the relevance of this systematic review, and the study is well writtenand well structured.
Thank You for the positive opinion and a kind comment.
The introduction is clear and adequate. This sentence could be improved or eliminated (lines 62-63). Itseems too conversational to me "Thus, the present study's authors have decided to find scientific
evidence to answer the question". It is better to rephrase the whole aims paragraph,
putting the questions on lines 64 to 68 as the aim of the study.
Thank You for the positive opinion and kind suggestion. The fragment indicated has been corrected as follows:
Thus, the authors of the present study aimed to find scientific evidence referring to the use of attachments, especially the way of their optimal placement for efficient tooth movement, as well as their number and shape. The aim of this systematic review is to comprehensively discuss the scientific evidence on how composite attachments should be bonded to achieve the best possible treatment results in aligner therapy.
The materials and methods section is ok.
Thank You for the positive opinion and a kind comment.
The results and tables are well presented.
Thank You
Discussion:
- 4.1 Section is ok.
Thank You
- 4.2 It should be better explained why palatal attachments are more effective in some (not
all) movements. There are some biomechanical reasons.
Thank You for this remark. The authors have added the following sentence: It should be kept in mind that palatal attachments have a different biomechanical effects, as positioned closer to the centre of resistance.
I have doubts about the table. Since different types of attachments were described in the studies,
on what basis did you choose the most effective one?
Thank You for Your kind question. The authors would like to explain that the attachments described in the literature as most effective have been placed in the table. A suitable sentence has been to text for explanation in order address this kind comment.
In general, the last paragraph of this section is very confusing and does not give appropriate
consideration of the predictability findings in the studies included in this review and the
literature.
Thank You for valuable suggestion. The last paragraph of the discussion has been revised as follows:
Possible limitations of the present study may be due to the fact, that the published clinical studies are based on data collected in different periods of time, thus are mixing together treatments performed according to different aligner protocols or are based on old protocols - like the study published in 2019 basing on Invisalign G6 protocol. By the time of writing this manuscript, the current Invisalign treatment protocol is 8/8+ and scientific evidence about the improvement between the protocols in purely descriptive [49], without any clinical trials, assessing the effect of specific types of movements. The authors of the study cited indicate that overcorrection should still be used. The impact of currently used aligner protocols on the treatment results has no scientific evidence, yet.
There are case series published in the orthodontic literature, pointing to the advantages a hybrid treatment, including the use of aligners with TAD supported anchorage devices, tubes or segmental orthodontic appliance (3-4 brackets) [50,51,52]. In a recently published handbook for aligner treatment, such a procedure is often even advised [53]. However, no studies provide scientific background for aligner only or hybrid treatment.
References are wrong; please correct them.
Thank You for Your remark. The references has been corrected.
There are some typos in the text.
Thank You, we have cross-checked and corrected the errors.
Thank you.
My best Regards
The authors hope that the manuscript in its present form will meet Your requirements.
Reviewer 3 Report
In the present study, Authors performed a systematic review on the shape, placement and bonding of composite attachments to achieve the best treatment results in aligner therapy.
The topic is original, and the research design is appropriate. Moreover, the structure of the paper is clearly presented and well-documented. Figures and tables are relevant to the research, and the discussions and conclusions are well-presented. Overall, the manuscript appears to provide a nice contribution to the literature. However, the Authors should improve the introduction section and revise the references because there is no number match between the articles cited in the text and the Reference section.
Author Response
Dear Reviewer,
Please find the answers to the Your comments in black italics.
The revised manuscript includes all the changes made shown in red type.
In the present study, Authors performed a systematic review on the shape, placement and bonding of composite attachments to achieve the best treatment results in aligner therapy.
Thank You for Your positive reception of manuscript.
The topic is original, and the research design is appropriate. Moreover, the structure of the paper is clearly presented and well-documented. Figures and tables are relevant to the research, and the discussions and conclusions are well-presented. Overall, the manuscript appears to provide a nice contribution to the literature. However, the Authors should improve the introduction section and revise the references because there is no number match between the articles cited in the text and the Reference section.
Thank You for detailed reviewing and suggestions made. The introduction and references has been corrected as kindly recommended.
The authors hope that the manuscript in its present form will meet Your requirements.
Reviewer 4 Report
The author filtrated and analyzed 26 latest and representative articles, which helps solve some key questions in the field of the attachments for the orthodontic aligner treatment and put forward limitations at present. The filtering strategy is reasonable and the conclusion is comprehensive. Overall, the article is well organized and its presentation is good. However, some minor issues still need to be improved.
1. The title seems to be kind of over-scaled. The paper just discusses a small part of the “attachments for the orthodontic aligner treatment”, which does not match the title.
2. The first paragraph of the Introduction is confused. Why the author began with “invisible thermoformed splints” whose necessity is doubtful.
3. The Introduction only involves solely use and composite use of attachment and then raises numerous other questions, which is lack of logicality. The Introduction should comprehensively discuss the background of your paper and logically lead to the questions.
4. Simplification and classification of the conclusion of the 26 articles could be more systematical.
5. There are a few typos and grammar errors in this paper.
Author Response
The author filtrated and analyzed 26 latest and representative articles, which helps solve some key questions in the field of the attachments for the orthodontic aligner treatment and put forward limitations at present. The filtering strategy is reasonable and the conclusion is comprehensive. Overall, the article is well organized and its presentation is good. However, some minor issues still need to be improved.
Thank You for Your positive opinion and positive comments
- The title seems to be kind of over-scaled. The paper just discusses a small part of the “attachments for the orthodontic aligner treatment”, which does not match the title.
Dear reviewer, the authors are open to suggestions, however we do not understand what kind of correction is proposed. The authors do not consider the study a “small part” as proper placement of composite attachments, proper shape of the attachments to achieve planned tooth movement, proper number of attachments on adjacent teeth for maintaining proper mutual anchorage, proper materials for attachment bonding and characteristics proper transfer splint were covered. The authors do not know, what other issues could have been searched in this regard.
- The first paragraph of the Introduction is confused. Why the author began with “invisible thermoformed splints” whose necessity is doubtful.
Dear reviewer, as suggested, the first paragraph has been removed.
- The Introduction only involves solely use and composite use of attachment and then raises numerous other questions, which is lack of logicality. The Introduction should comprehensively discuss the background of your paper and logically lead to the questions.
The last paragraph of introduction has been revised for a better clarity, as proposed.
The fragment indicated has been corrected as follows:
Thus, the authors of the present study aimed to find scientific evidence referring to the use of attachments, especially the way of their optimal placement for efficient tooth movement, as well as their number and shape. The aim of this systematic review is to comprehensively discuss the scientific evidence on how composite attachments should be bonded to achieve the best possible treatment results in aligner therapy.
- Simplification and classification of the conclusion of the 26 articles could be more systematical.
Thank You for this kind suggestion. The conclusions have been corrected as follows:
- The presence of attachments significantly improves aligner retention and orthodontic tooth movement
- Conventional attachments, with at least one beveled edge, are beneficial both for tooth movement and for anchorage. On anterior teeth, it is worth placing attachments on the lingual surfaces, both for therapeutic and aesthetic indications.
- It seems reasonable to recommend liquid composites or orthodontic bonding composites with as much filler as possible bonded with rigid transfer tray for better accuracy and composite bond strength.
- Future studies could concentrate on clinical tooth movement assessment in aligner therapy, efficacy of new generations of aligner treatment protocols and eventual indications for “hybrid treatment”
The authors hope that the manuscript in its present form will meet Your requirements.
Round 2
Reviewer 1 Report
Thank you for addressing the comments